# Caveolar disruption causes contraction of rat femoral arteries via reduced basal NO release and subsequent closure of BK$_{Ca}$ channels

AY Al-Brakati[1], T Kamishima[1], C Dart[2] and JM Quayle[1]

[1] Department of Cellular and Molecular Physiology, Institute of Translational Medicine, University of Liverpool, Liverpool, UK
[2] Department of Biochemistry and Cell Biology, Institute of Integrative Biology, University of Liverpool, Liverpool, UK

Corresponding author
JM Quayle, JQuayle@liv.ac.uk

## ABSTRACT

**Background and Purpose.** Caveolae act as signalling hubs in endothelial and smooth muscle cells. Caveolar disruption by the membrane cholesterol depleting agent methyl-$\beta$-cyclodextrin (M-$\beta$-CD) has various functional effects on arteries including (i) impairment of endothelium-dependent relaxation, and (ii) alteration of smooth muscle cell (SMC) contraction independently of the endothelium. The aim of this study was to explore the effects of M-$\beta$-CD on rat femoral arteries.

**Methods.** Isometric force was measured in rat femoral arteries stimulated to contract with a solution containing 20 mM K$^+$ and 200 nM Bay K 8644 (20 K/Bay K) or with one containing 80 mM K$^+$ (80 K).

**Results.** Incubation of arteries with M-$\beta$-CD (5 mM, 60 min) increased force in response to 20 K/Bay K but not that induced by 80 K. Application of cholesterol saturated M-$\beta$-CD (Ch-MCD, 5 mM, 50 min) reversed the effects of M-$\beta$-CD. After mechanical removal of endothelial cells M-$\beta$-CD caused only a small enhancement of contractions to 20 K/Bay K. This result suggests M-$\beta$-CD acts via altering release of an endothelial-derived vasodilator or vasoconstrictor. When nitric oxide synthase was blocked by pre-incubation of arteries with L-NAME (250 μM) the contraction of arteries to 20 K/Bay K was enhanced, and this effect was abolished by pre-treatment with M-$\beta$-CD. This suggests M-$\beta$-CD is inhibiting endothelial NO release. Inhibition of large conductance voltage- and Ca$^{2+}$-activated (BK$_{Ca}$) channels with 2 mM TEA$^+$ or 100 nM Iberiotoxin (IbTX) enhanced 20 K/Bay K contractions. L-NAME attenuated the contractile effect of IbTX, as did endothelial removal.

**Conclusions.** Our results suggest caveolar disruption results in decreased release of endothelial-derived nitric oxide in rat femoral artery, resulting in a reduced contribution of BK$_{Ca}$ channels to the smooth muscle cell membrane potential, causing depolarisation and contraction.

**Peer**J ___________________________________

## INTRODUCTION

Caveolae are flask shaped invaginations, 50–100 nm in diameter, present in the plasma membrane of endothelial cells (ECs) and smooth muscle cells (SMCs) (*Cohen et al., 2004*). Caveolae have been implicated in cell signalling and transport processes in the vasculature, with roles that include the regulation of contractility, trans-endothelial transport and cholesterol metabolism (*Cohen et al., 2004*). Many of the functions of caveolae are attributed to the caveolin family of proteins, of which cavolin-1 is the most widely expressed subtype (*Cohen et al., 2004*).

Disruption of caveolae causes arterial contractility to change by acting on both ECs and SMCs. Disruption of endothelial caveolae by cholesterol depleting agents such as M-$\beta$-CD and filipin reduces responses to endothelial-dependent vasodilators (*Darblade et al., 2001*; *Xu et al., 2008*). Caveolin-1 has a tonic inhibitory effect on endothelial NO synthase (eNOS) activity, and it is thought that on cholesterol-depletion caveolin-1 re-distributes from caveolar to non-caveolar membranes (*Xu et al., 2008*). This results in a reduction in eNOS activation in response to endothelium-dependent vasodilators, possibly due to disrupted organisation of signalling complexes normally located in caveolae (*Xu et al., 2008*). As well as regulating NO synthesis, caveolae are implicated in the response to other endothelium-derived vasodilators including endothelial derived hyperpolarising factor (EDHF) (*Graziani et al., 2004*; *Xu et al., 2007*). Caveolar disruption may underlie the reduced endothelium-dependent vasodilation seen in a number of diseases, including atherosclerosis (*Darblade et al., 2001*; *Xu et al., 2008*).

Caveolae in SMCs also regulate contractility (e.g., *Razani et al., 1990*; *Dreja et al., 2002*; *Potoknik et al., 2007*). The response to some vasoconstrictors (e.g., 5-hydroxytryptamine, endothelin-1) is reduced after caveolar disruption (*Dreja et al., 2002*; *Prendergast et al., 2010*), although the response to elevated extracellular $K^+$ and to $\alpha$-adrenergic agonists is generally unaffected (*Dreja et al., 2002*; *Prendergast et al., 2010*; though see *Je et al., 2004*). Myogenic tone that develops in response to pressurization of small resistance-sized arteries is reduced (*Dubroca et al., 2007*; *Potoknik et al., 2007*). Caveolar disruption may lead to a reduced responsiveness to vasoconstrictors and pressure via loss of caveolar localisation of signal transduction pathways (e.g., *Clarke, Ohanian & Ohanian, 2007*; *Dubroca et al., 2007*), ion channels (e.g., *Sampson et al., 2007*) and $Ca^{2+}$-signalling molecules (e.g., *Löhn et al., 2000*; *Shaw et al., 2006*).

One important influence on vascular tone is the activity of large conductance $Ca^{2+}$- and voltage-activated $K^+$ ($BK_{Ca}$) channels (*Wray & Burdyga, 2010*). Indeed, $BK_{Ca}$ channel dysfunction occurs in several vascular diseases including hypertension and atherosclerosis, and pharmacological modulation of these channels is a promising avenue for novel treatments (*Félétou, 2009*). $BK_{Ca}$ channel regulation is complex and multi-factorial, with influences including $Ca^{2+}$, voltage, cellular microarchitecture, protein kinases, and the accessory $\beta 1$- subunit which enhances the $Ca^{2+}$ sensitivity of the pore forming $\alpha$ subunit (reviewed by *Hill et al., 2010*). As a consequence of this complex regulation, $BK_{Ca}$ channels show artery and vascular bed-specific properties (*Hill et al., 2010*). In cerebral resistance arteries, SR $Ca^{2+}$ release through ryanodine receptors (Ry-R) triggers $Ca^{2+}$ 'sparks' in

SMCs (*Nelson et al., 1995*). The resulting local increase in $[Ca^{2+}]$ in the sub-sarcolemmal domain triggers activation of clusters of $BK_{Ca}$ channels in the plasma membrane, resulting in spontaneous transient outward currents (STOCs). These STOCs result in membrane hyperpolarisation, closure of VDCCs and relaxation, providing a brake on myogenic tone (*Nelson et al., 1995*). It is thought that caveolae allow the plasma membrane to come into closer opposition with the superficial SR and so localised $Ca^{2+}$ signalling between SR and $BK_{Ca}$ channels can occur in the sub-membrane domain (*Löhn et al., 2000*; *Drab et al., 2001*; *Wray & Burdyga, 2010*). Caveolins bind directly to $BK_{Ca}$ channels and this likely serves to localise them within caveolae (e.g., *Yamamura et al., 2012*). In contrast to cerebral resistance arteries, some other arteries do not display this tight spark-STOC coupling. In particular, arteries supplying skeletal muscle have $BK_{Ca}$ channels that have lower $Ca^{2+}$ sensitivity and that are not activated by sparks, and this is thought to be due to lower expression of the $\beta 1$ subunit relative to the $\alpha$ subunit as well as a lower overall level of $\alpha$-subunit expression (*Yang et al., 2009*; *Yang et al., 2013*; *Nourian et al., 2014*). Decreased activity results in a smaller contribution of $BK_{Ca}$ channels to the cell membrane potential and this may in turn permit higher levels of vascular resistance in the skeletal muscle circulation (*Jackson & Blair, 1998*; *Hill et al., 2010*).

The contribution of caveolae and $BK_{Ca}$ channels to arterial contractility in response to caveolar disruption are complex and artery dependent. Furthermore, a role for endothelial-derived substances or endothelial caveolae on SMC $BK_{Ca}$ channel activity has not been reported. In this study, we have investigated the effects of caveolar disruption by membrane cholesterol depletion on the contraction of rat femoral artery *in vitro*. Our results are consistent with caveolar loss causing reduced NO release from ECs. Lower NO availability results in a decrease in the contribution of $BK_{Ca}$ channels to the SMC membrane potential, leading to depolarisation and smooth muscle cell contraction. Caveolar disruption has only a small direct effect on SMCs in the absence of the endothelium in rat femoral arteries.

# MATERIALS AND METHODS

## Animals

Tissues were obtained from adult male Wistar rats (175–200 g; Charles River Laboratories) which were killed by a rising concentration of $CO_2$ followed by exsanguination. The care and euthanasia of animals conformed to the requirements of the UK Animals (Scientific Procedures) Act 1986.

## Myography

Isometric tension of rat femoral arteries was measured in a small artery myograph (Model 500A; Danish Myotechnology, Aarhus, Denmark) as previously described (*Quayle et al., 2006*). Arteries were dissected in cold 5.4 mM $K^+$ physiological saline solution (PSS) containing (mM): 137 NaCl, 5.4 KCl, 0.44 $NaH_2PO_4$, 0.42 $Na_2HPO_4$, 4.17 $NaHCO_3$, 1 $MgCl_2$, 2 $CaCl_2$, 10 HEPES, 10 glucose, pH adjusted to 7.4 with NaOH. Ring segments of artery were then mounted in the myograph by threading two strands of tungsten wire

(diameter 40 μm) through the vessel lumen. After vessel mounting and equilibration at 37 °C, passive tension was adjusted to allow measurement of active force production (*Mulvany & Halpern, 1977*). Arteries were stimulated to contract by; (i) an 80 mM $K^+$ containing saline (80 K), which had the composition of the 5.4 mM $K^+$ saline except 74.6 mM NaCl was substituted with KCl, or, (ii) a 20 K/Bay K containing saline, which had the composition of the 5.4 mM $K^+$ saline except 14.6 mM NaCl was substituted with KCl and that 200 nM of the dihydropyridine voltage-dependent calcium channel agonist ±Bay K8644 had been added. This solution has proved useful for inducing arterial contraction in the absence of agonists that are coupled to receptors and second messenger pathways (*Davie, Kubo & Standen, 2008*). The endothelium was removed from some arteries by rubbing a human hair through the lumen. Removal of a functional endothelium was confirmed by absence of relaxation to 10 μM acetylcholine.

## Transmission electron microscopy

Femoral arteries were treated with 5 mM M-$\beta$-CD ($n = 5$) and matched with a control group of arteries ($n = 5$). Both groups were cut into small pieces (about 0.5 mm$^3$). The segments were immediately fixed with 4% paraformaldehyde and 2.5% glutaraldehyde in 0.1 M sodium cacodylate (pH 7.4) overnight at room temperature. Next day, the samples were rinsed three times in 0.1 M sodium cacodylate and then post-fixed in 1% (w/v) osmium tetroxide (OsO$_4$) in 0.1 M sodium cacodylate for 1 h. The samples were then rinsed with 0.1 M cacodylate for 30 min, and incubated with 0.1 M cacodylate overnight. Samples were washed with distilled water and ethanol, 30 min each, and incubated in 2% aqueous uranyl acetate for 60 min before embedding in resin. Samples were dehydrated through a graded series of alcohol (60, 70, 80, 90 and 100%), five minutes each. Segments were then immersed in 100% acetone to remove water and then embedded in resin (30% resin:70% acetone; 70% resin:30% acetone and 100% resin) for 1, 1 and 2 h, respectively. Samples were left in the oven at 60 °C to polymerise overnight. Ten sections were cut from each group (i.e., control and M-$\beta$-CD treated) at a thickness of 70–90 nm using an ultramicrotome (Leica EM FC6). Ultrathin sections were picked up on coated copper grids of 300-mesh, with 0.3% Pioloform and the grids dried overnight at room temperature. Grids were then double stained with 2% aqueous uranyl acetate for four minutes, and thereafter rinsed with distilled water for a minute, re-stained again with 0.1 M lead citrate for four minutes, washed with distilled water for a minute and left overnight at room temperature to dry.

## Drugs and chemicals

5 mM M-$\beta$-CD was dissolved directly in the extracellular solution. 5 mM cholesterol-saturated M-$\beta$-CD (Ch-MCD) was prepared by dissolving directly in PSS by heating at 80 °C for 10 min using a water bath. Filipin (4 μg/ml) was prepared by dissolving in PSS at 37 °C for 10 min using a water bath. L-NAME (250 μM) was dissolved directly in PSS. ±BayK 8644 was made up as a 5 mM stock in ethanol and added to the 20 K extracellular solution. Tetraethylammonium (TEA$^+$) chloride and iberiotoxin (IbTX) were dissolved in the appropriate extracellular solution.

## Data analysis and statistics

Data were recorded on computer using a MiniDigi analogue to digital interface in combination with Axoscope software (Axon Instruments, Union City, California, USA). Results were analysed using Axoscope and SigmaPlot 11. The passive or resting tension was subtracted prior to data analysis, and active force measured in the steady state is shown. Statistical significance was assessed by ANOVA with Tukey's post-test comparison or by paired Student's $t$-test, using GraphPad InStat 3 software. Significance is given as $P < 0.05$(*), $P < 0.01$(**), $P < 0.001$(***) or not significant (n.s.) Data are reported as mean $\pm$ SEM, and $n$ is the number of arteries, isolated from at least three animals.

# RESULTS

## Cholesterol depletion by M-$\beta$-CD disrupts caveolae in smooth muscle and endothelial cells of rat femoral artery

M-$\beta$-CD depletes membrane cholesterol, which is essential for caveolar stability (*Zidovetzki & Levitan, 2007*). Figure 1 illustrates transmission electron micrographs of sections of control (untreated) and M-$\beta$-CD treated rat femoral artery. Omega shaped invaginations of the plasma membrane characteristic of caveolae were seen in control sections in both ECs and SMCs. After treatment with M-$\beta$-CD, caveolae were seen less frequently, and those that were present showed an opened-out, cup-like profile. These images establish that treatment with M-$\beta$-CD disrupts caveolae of rat femoral ECs and SMCs.

## Disruption of caveolae enhances femoral artery contraction

To investigate the functional role of caveolae in rat femoral artery, the effect of treatment with M-$\beta$-CD on the contractile response to 80 K and 20 K/Bay K was recorded (Fig. 2). Both solutions will cause contraction by membrane depolarisation and activation of voltage-dependent calcium channels in SMCs (e.g., *Meisheri, Cipkus Dubray & Oleynek, 1990*; *Davie, Kubo & Standen, 2008*). After M-$\beta$-CD treatment, 20 K/Bay K-induced contraction significantly increased from $11.84 \pm 1.30$ mN to $18.25 \pm 2.00$ mN ($n = 12$, $P < 0.01$). In contrast, the contraction to 80 K was virtually identical, with a maximum force of $33.40 \pm 1.43$ mN before and $33.16 \pm 2.02$ mN after M-$\beta$-CD treatment ($n = 12$, n.s.). Filipin, a macrolide antibiotic which binds cholesterol and disrupts caveoli, also significantly increased force in response to 20 K/Bay K (Fig. 3) ($5.36 \pm 1.66$ mN to $9.36 \pm 2.21$ mN, $n = 14$, $P < 0.001$), but not 80 K ($18.60 \pm 2.92$ to $18.73 \pm 3.12$, n.s.)

M-$\beta$-CD might have non-specific effects (*Zidovetzki & Levitan, 2007*). To examine if the above results were due to caveolae disruption, we attempted to reverse the effects of M-$\beta$-CD by subsequent incubation of arteries with cholesterol saturated M-$\beta$-CD (Ch-MCD) (e.g., *Dreja et al., 2002*; *Prendergast et al., 2010*). Ch-MCD (5 mM, 50 min) had no significant effect on 20 K/Bay K induced tone by itself ($3.15 \pm 0.45$ mN to $2.98 \pm 0.59$ mN, $n = 6$, n.s.). However, Ch-MCD reversed the enhancing effect of M-$\beta$-CD on 20 K/Bay K contractions (Fig. 4), with force changing from $3.50 \pm 0.61$ mN in 20 K/Bay K, to $5.60 \pm 1.46$ mN after M-$\beta$-CD, to $3.02 \pm 0.64$ mN following Ch-MCD ($n = 14$). Ch-MCD significant inhibited contractions to 80 K solution, with force changing

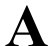

# A

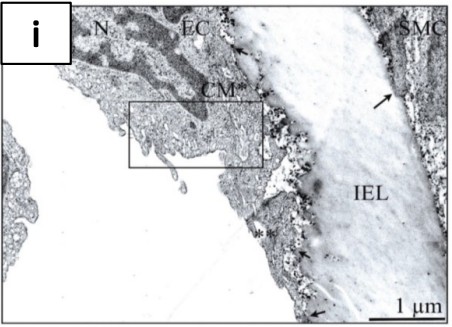

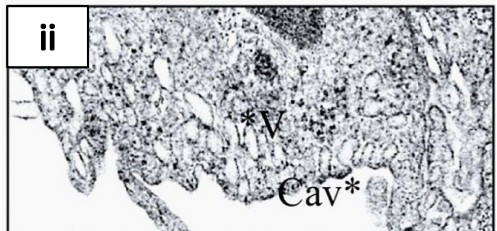

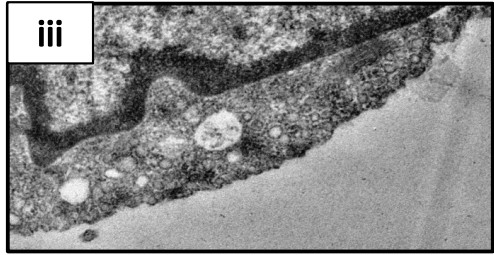

# B

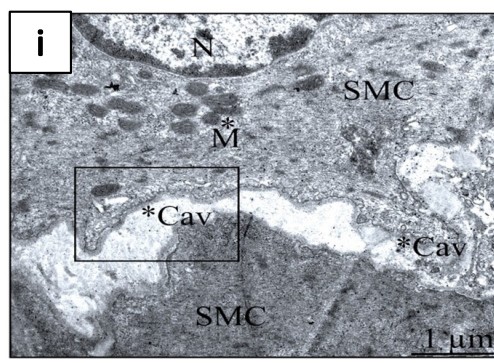

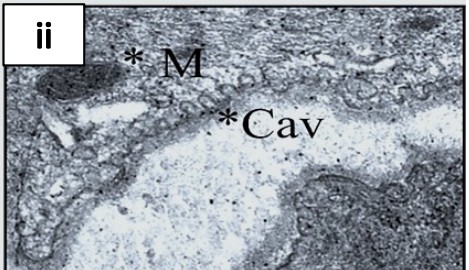

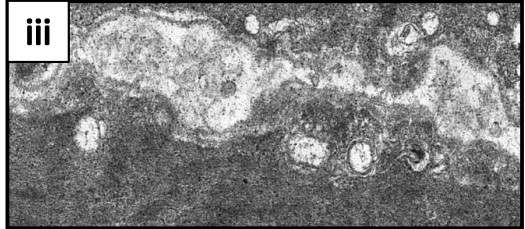

**Figure 1 M-β-CD disrupts caveolae in rat femoral artery.** (A) (i) Transmission electron micrograph of rat femoral artery showing caveolae covering most of the EC membrane. IEL is the internal elastic lamina. Magnification was ×60,000. Scale bar = 1 μm. (ii) Detail from marked area in (i) showing caveolae (Cav) and caveosomes (V). (iii) After treatment with 5 mM M-β-CD, EC cell membrane was free of caveolae. (B) (i) TEM of smooth muscle cell. Magnification was ×60,000. Scale bar = 1 μm. (ii) Detail from marked area in (i) showing caveolae (Cav) and mitochondrion (M). (iii) After treatment with 5 mM M-β-CD, SMC cell membrane was free of caveolae.

**A**

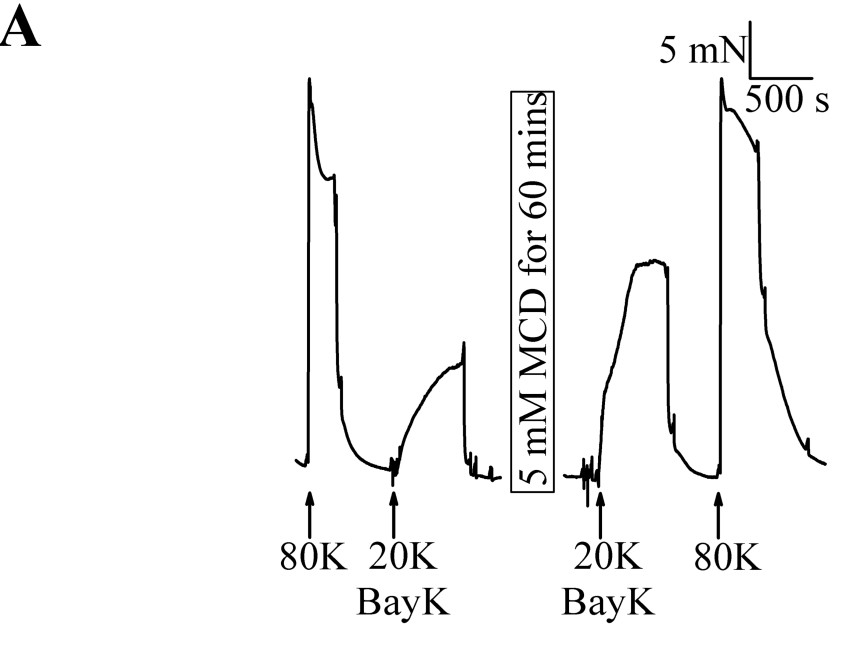

**B**

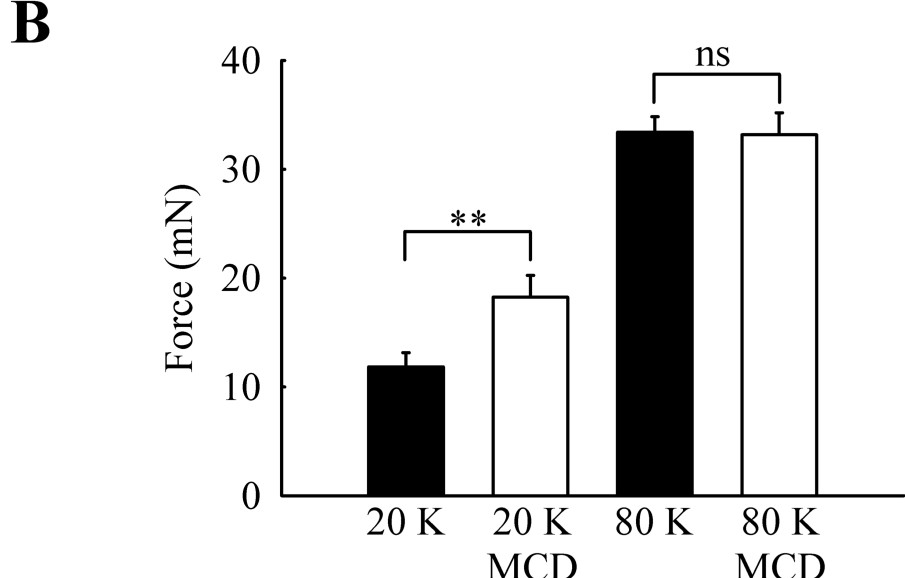

**Figure 2 Effect of caveolar disruption with M-β-CD on rat femoral artery contraction.** (A) Original traces show that treatment with 5 mM M-β-CD augments contraction in response to 20 K/Bay K but not to 80 K. (B) Mean data showing the effects of cholesterol depletion on the response to 20 K/Bay K and 80 K. Statistically significant difference was detected using Student's $t$-test. $n = 12$.

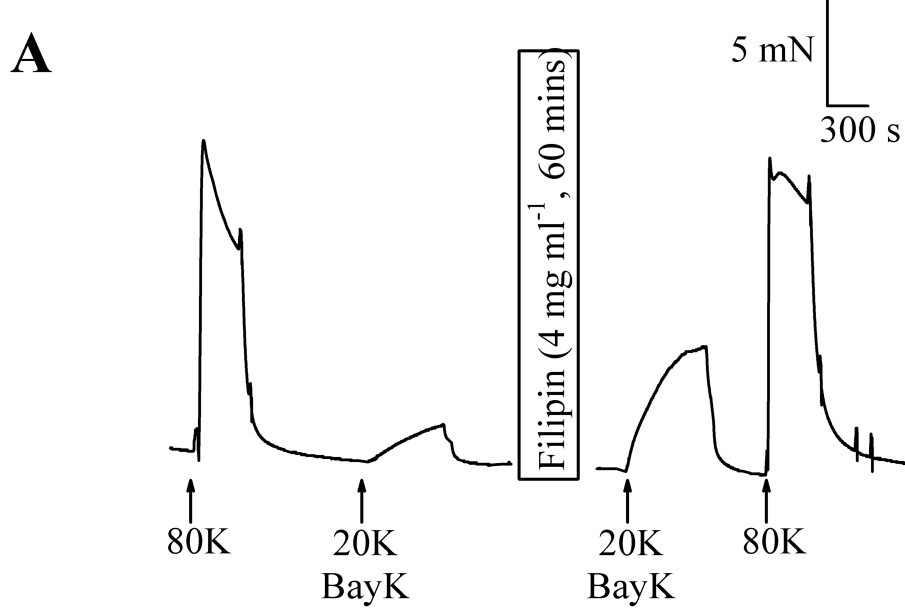

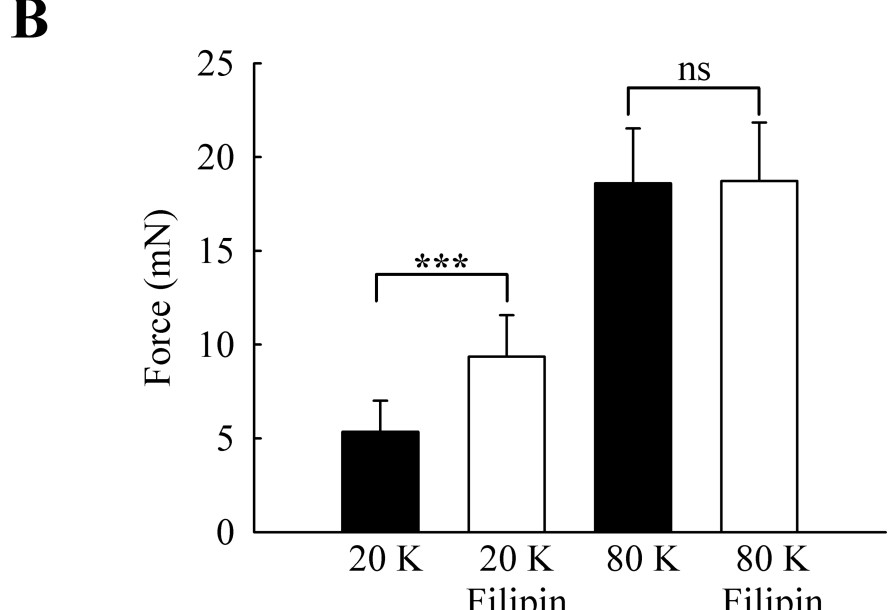

**Figure 3 Effect of caveolar disruption with filipin.** (A) Filipin treatment augments contraction in response to 20 K/Bay K but not to 80 K. (B) Mean data showing the effects of filipin treatment on the response to 20 K/Bay K and 80 K. Statistical significance was detected using Student's $t$-test. $n = 14$.

**A**

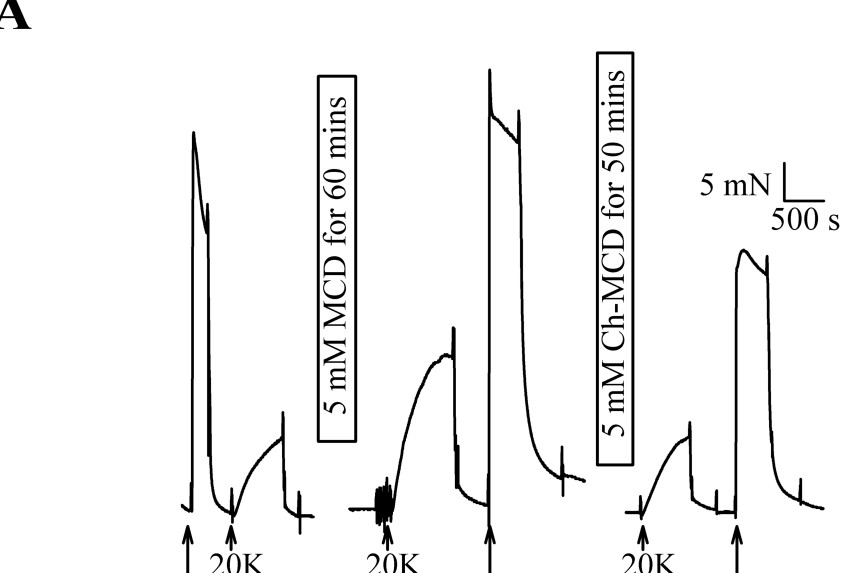

**B**

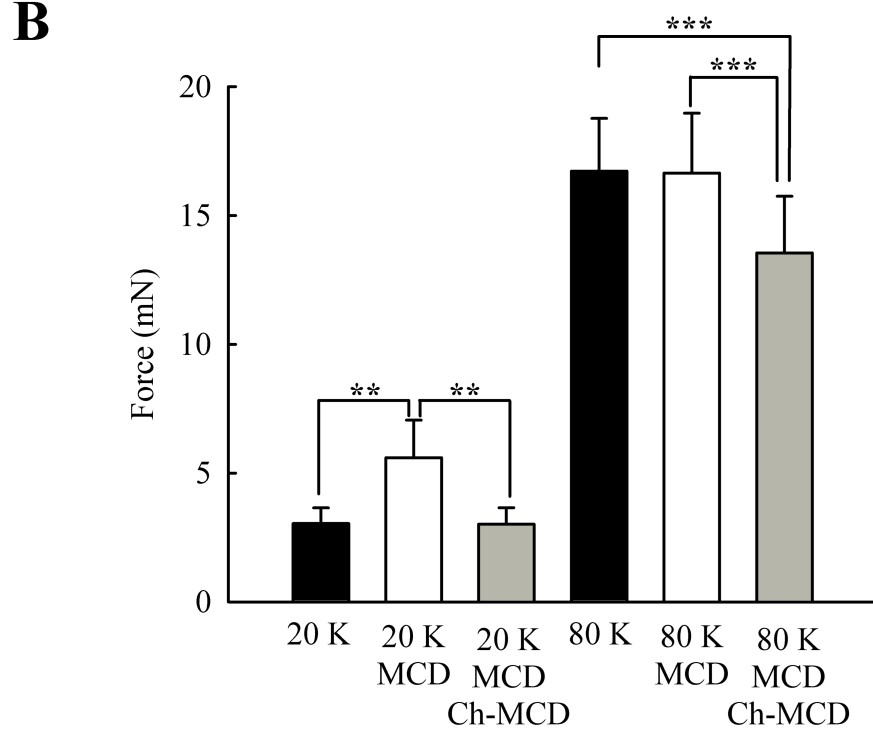

**Figure 4 Ch-MCD reverses the effect of M-β-CD on contractions.** (A) M-β-CD treatment augments contraction to 20 K/ BayK and Ch-MCD reverses this effect. (B) Mean data showing the effect of M-β-CD and Ch-MCD on 20 K/Bay K and 80 K contractions. Statistical significance was examined using ANOVA. $n = 14$.

from $16.72 \pm 2.05$ mN in 80 K to $13.55 \pm 2.20$ mN in 80 K in the presence of Ch-MCD ($n = 14, P < 0.001$).

### The effect of M-$\beta$-CD on force is endothelium-dependent

Caveolar disruption has been reported to reduce endothelium-dependent vasodilation but it can also alter contraction by directly acting on SMCs (see Introduction). In order to distinguish these two possibilities, ECs were mechanically removed and absence of a functional endothelium confirmed for each artery by a lack of a vasodilator response to 10 μM acetylcholine. Endothelial removal reduced the ability of M-$\beta$-CD to enhance contractions to 20 K/Bay K (Fig. 5) ($6.62 \pm 0.92$ mN to $7.52 \pm 1.03$ mN after M-$\beta$-CD, $n = 22$, n.s.). The results indicate that M-$\beta$-CD acts on endothelium where it alters release of an endothelial-derived vasodilator or vasoconstrictor.

### L-NAME enhances 20 K/Bay K contractions but has no effect after M-$\beta$-CD treatment

One candidate for an endothelial factor influencing SMC contraction is NO. When arteries were pre-incubated with the eNOS inhibitor L-NAME (250 μM) the contraction to 20 K/Bay K was enhanced (Fig. 6) ($6.82 \pm 1.61$ mN to $14.90 \pm 2.67$ mN, $n = 6, P < 0.001$). However, after M-$\beta$-CD treatment, L-NAME no longer had a significant effect on the response to 20 K/Bay K ($17.74 \pm 3.46$ mN to $16.10 \pm 3.19$ mN, $n = 6$, n.s). This result is consistent with the hypothesis that caveolar disruption by M-$\beta$-CD results in decreased release of NO from the endothelium.

### Effect of L-NAME on femoral artery responses to phenylephrine

Contraction induced with 20 K/Bay K has the advantage of being relatively simple in mechanism, by-passing receptors and second messenger systems. However, it may not reflect events occurring during more physiological contractions. To address whether basal release of NO occurred in the presence of a physiological stimulus the effect of L-NAME was examined in arteries contacted with the $\alpha$-adrenergic agonist, phenylephrine. Contraction was triggered with cumulative doses of phenylephrine (PE) (0.1–30 μM) (Fig. S1). Arteries were then pre-incubated with L-NAME (250 μM) for 10 min and phenylephrine applied in the presence of L-NAME. A given concentration of phenylephrine was more effective after L-NAME treatment, and concentration–response curves were shifted to the left (Fig. S1). This results suggests endothelial NO synthesis also occurs during phenylephrine contractions in these arteries.

### Effect of inhibition of BK$_{Ca}$ channels by TEA$^+$ and iberiotoxin on the response to M-$\beta$-CD

M-$\beta$-CD augmented the response to 20 K/Bay K but not to 80 K (see Fig. 2). In 80 K solution the cell membrane potential will lie close to the equilibrium potential for K$^+$. In this case, opening of K$^+$ channels will not lead to hyperpolarisation, and the inability of a substance to cause relaxation in 80 K solution may implicate K$^+$ channel activation in vasodilation (e.g., *Meisheri, Cipkus Dubray & Oleynek, 1990*). One explanation for our data is therefore that NO release from the endothelium causes SMC relaxation via K$^+$ channel ac-

**A**

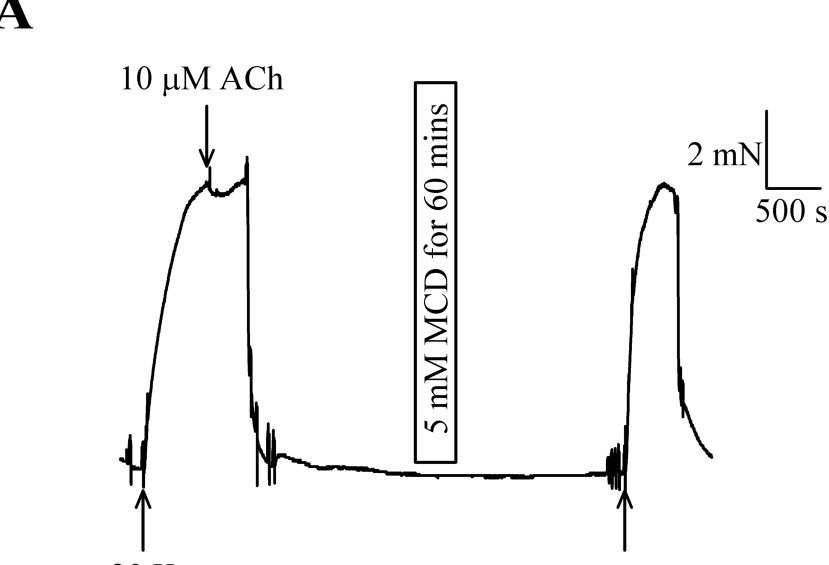

**B**

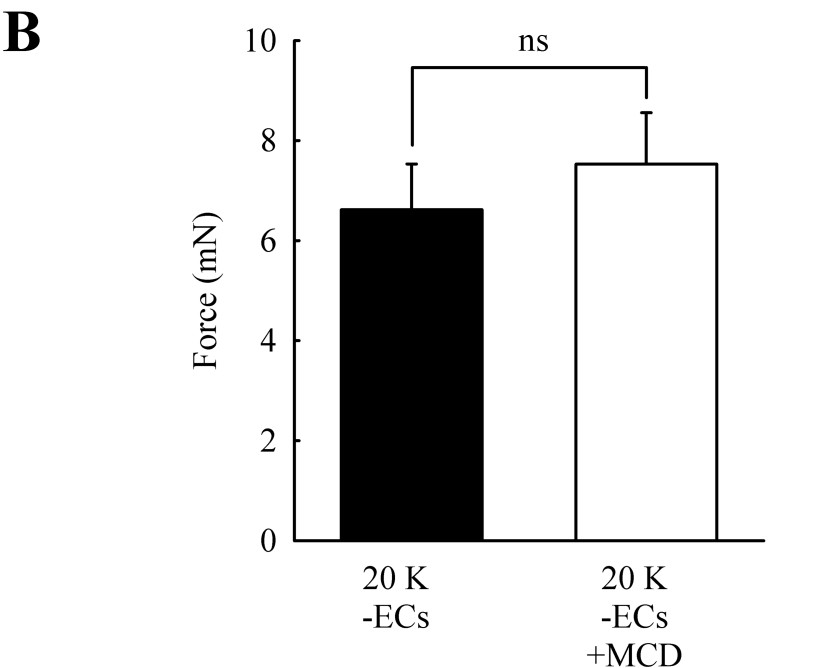

**Figure 5 Effect of M-$\beta$-CD on endothelium-denuded femoral artery.** (A) After endothelial removal incubation with M-$\beta$-CD did not augment contractions to 20 K/Bay K. Absence of a functional endothelium was shown by lack of relaxation to ACh. (B) Mean data showing no significant difference in the contractions of endothelium-denuded arteries stimulated with 20 K/Bay K following M-$\beta$-CD treatment. Statistical significance was examined using Student's $t$-test. $n = 22$.

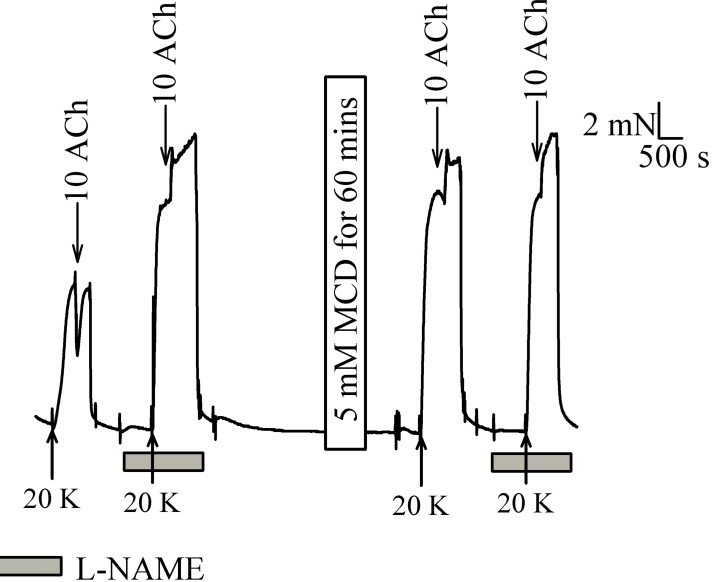

**A**

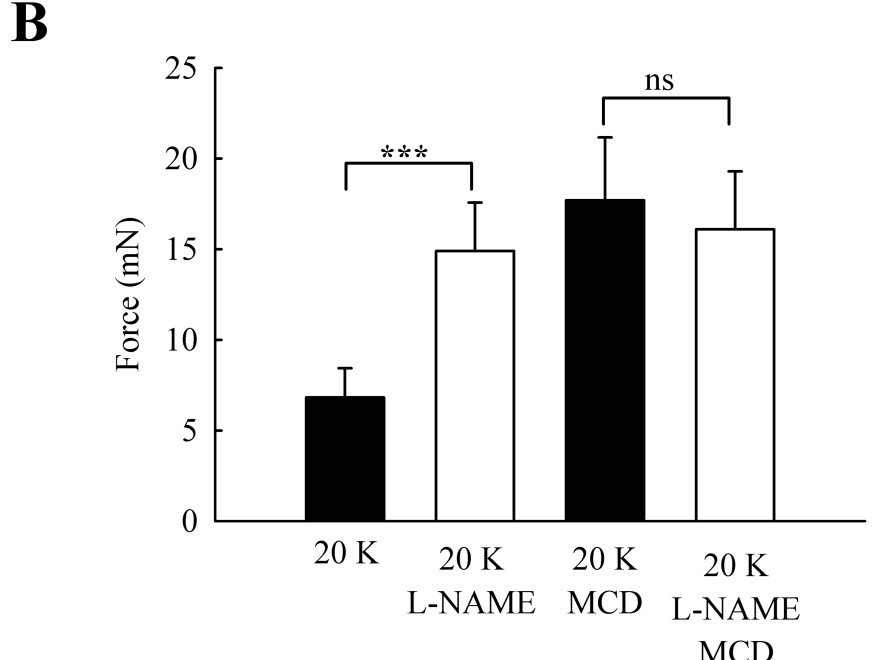

**B**

**Figure 6 Effect of inhibition of NO synthase by L-NAME.** (A) Incubation with 250 µM L-NAME augmented contraction response to 20 K/Bay K. When arteries were incubated with M-$\beta$-CD, contraction with 20 K/Bay K was enhanced. However, L-NAME no longer had an additional contractile effect after M-$\beta$-CD treatment. (B) Mean data showing 250 µM L-NAME significantly increases the contraction response to 20 K/Bay K. L-NAME did not enhance the contraction after M-$\beta$-CD treatment. Statistical significance was examined using ANOVA. $n = 6$.

tivation. In order to test this, the effect of the $BK_{Ca}$ channel inhibitors $TEA^+$ and IbTX was investigated before and after caveolar disruption with M-$\beta$-CD. $BK_{Ca}$ channels have been previously implicated in NO-induced vasodilation (e.g., *Kahn, Higdon & Meisheri, 1998*).

Initially, 2 mM $TEA^+$ was used to inhibit $BK_{Ca}$ channels, and at this concentration it can be regarded as selective (*Langton et al., 1991*). $TEA^+$ increased resting force from $1.15 \pm 0.68$ mN to $2.15 \pm 0.81$ mN ($n = 8$, ns, Fig. 7). In some experiments, including that illustrated in Fig. 7, spikes were superimposed on top of the increase in resting force, indicating $TEA^+$ is inducing spontaneous oscillations in this vessel. 2 mM $TEA^+$ application significantly increased the contraction with 20 K/Bay K from $7.07 \pm 2.05$ mN to $18.53 \pm 3.25$ mN ($n = 8$, $P < 0.05$). These results suggest $TEA^+$-sensitive $K^+$ channels contribute to the membrane potential at both resting state and during contraction with 20 K/Bay K. Incubating arteries with 5 mM M-$\beta$-CD augmented contraction to 20 K/Bay K, as previously shown ($7.07 \pm 2.05$ mN to $15.29 \pm 2.84$ mN; $n = 8$, $P < 0.01$). After treatment with M-$\beta$-CD, $TEA^+$ no longer caused significant additional contraction to 20 K/Bay K ($15.29 \pm 2.84$ mN to $19.54 \pm 2.22$ mN; $n = 8$, ns). The observation that $TEA^+$ significantly increased force before, but not after, M-$\beta$-CD treatment suggests caveolar disruption decreases the contribution of a $TEA^+$-sensitive $K^+$ channel to the membrane potential.

In endothelium-denuded arteries, 2 mM $TEA^+$ caused a small but non-significant increase in 20 K/Bay K contraction (from $6.06 \pm 1.44$ mN to $8.01 \pm 1.58$ mN; $n = 8$, ns, Fig. 8). M-$\beta$-CD application had no significant effect on 20 K/Bay K contraction in endothelium-denuded arteries, as previously shown ($6.06 \pm 1.44$ mN to $6.19 \pm 1.32$ mN; $n = 8$, ns). After endothelium-denuded arteries were incubated with M-$\beta$-CD, $TEA^+$ caused a small, non-significant, additional contraction in response to 20 K/Bay K, from $6.19 \pm 1.32$ mN to $7.23 \pm 1.61$ mN ($n = 8$, ns).

The experiments with $TEA^+$ were repeated with the more selective $BK_{Ca}$ channel inhibitor iberiotoxin (IbTX) (Fig. 9). Incubating arteries with 100 nM IbTX augmented contraction to 20 K/Bay K ($5.13 \pm 0.93$ mN to $11.60 \pm 1.84$ mN; $n = 6$, $P < 0.01$). After endothelial removal, incubation of arteries with 100 nM IbTX no longer significantly enhanced 20 K/Bay K contractions ($6.76 \pm 0.88$ mN to $8.30 \pm 1.15$ mN, $n = 6$, ns).

Overall, the results with $TEA^+$ and IbTX indicate that inhibition of $BK_{Ca}$ channels increases contraction to 20 K/Bay K in rat femoral artery. Endothelium removal largely abolishes this effect, as does caveolar disruption by treatment with M-$\beta$-CD. Both M-$\beta$-CD treatment and removal of endothelium may inhibit endothelial NO release. NO is a known activator of $BK_{Ca}$ channels in SMCs, so reduced release may decrease $BK_{Ca}$ channel activity, leading to SMC depolarization and contraction.

## Effects of IbTX on tone are reduced by inhibition of basal NO release

Further experiments were designed to examine whether the effects of L-NAME and IbTX on contraction were additive, an indication that two mechanisms act through different pathways. Adding IbTX (100 nM) at the peak of the 20 K/Bay K contraction induced significant further contraction (Fig. 10). When arteries were pre-treated with L-NAME, the 20 K/Bay K contractions were enhanced, again consistent with previous observations, but

**A**

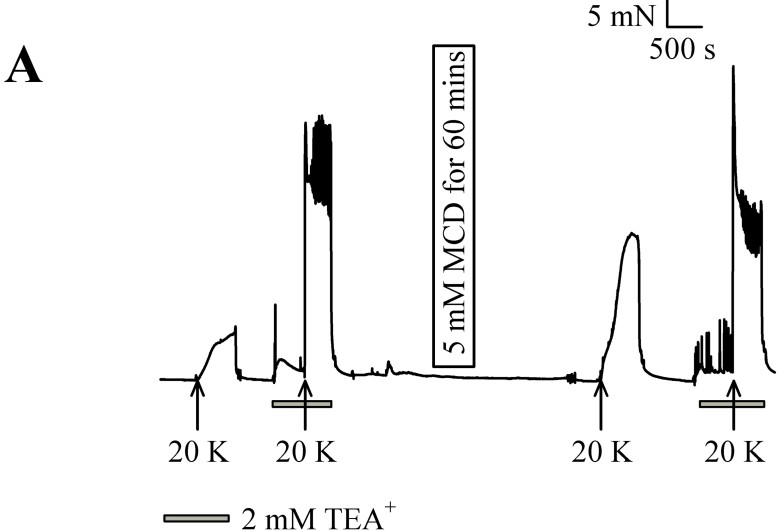

5 mN | 500 s

5 mM MCD for 60 mins

20 K          20 K                    20 K          20 K

2 mM TEA$^+$

**B**

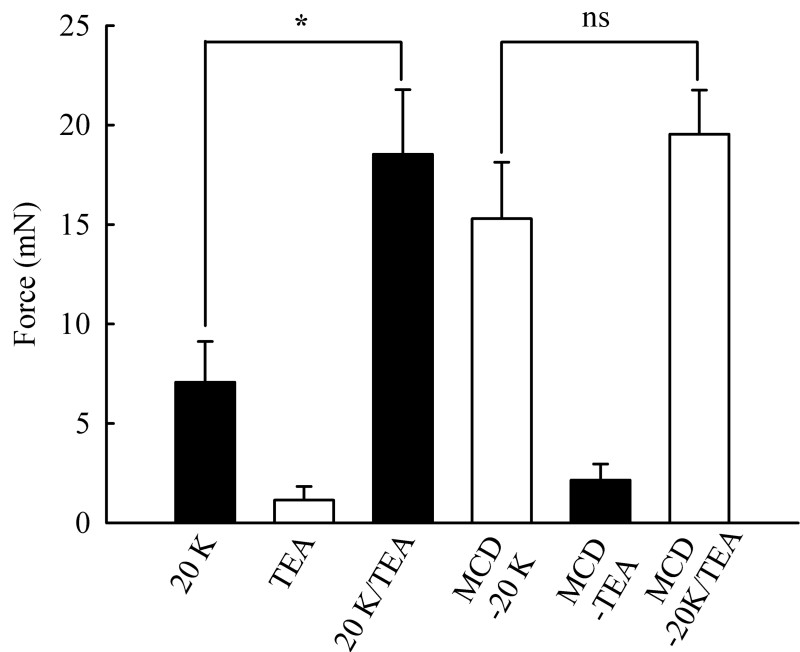

Figure 7 **Effect of TEA$^+$ on 20 K/Bay K contractions.** (A) Incubation of arteries with 2 mM TEA$^+$ increased basal tone, and also augmented the response to 20 K/Bay K. M-$\beta$-CD caused an enhanced contraction in response to 20 K/Bay K, but this effect of M-$\beta$-CD was absent in the presence of TEA$^+$. (B). Mean data for effect of TEA$^+$ on femoral artery contractions to 20 K/Bay K before and after M-$\beta$-CD treatment. Data show a significant increase in the contraction in response to 20 K/Bay K after incubation of the artery with 2 mM TEA$^+$ before, but not after, treatment with M-$\beta$-CD. Statistical significance was examined using ANOVA. $n = 8$.

**A**

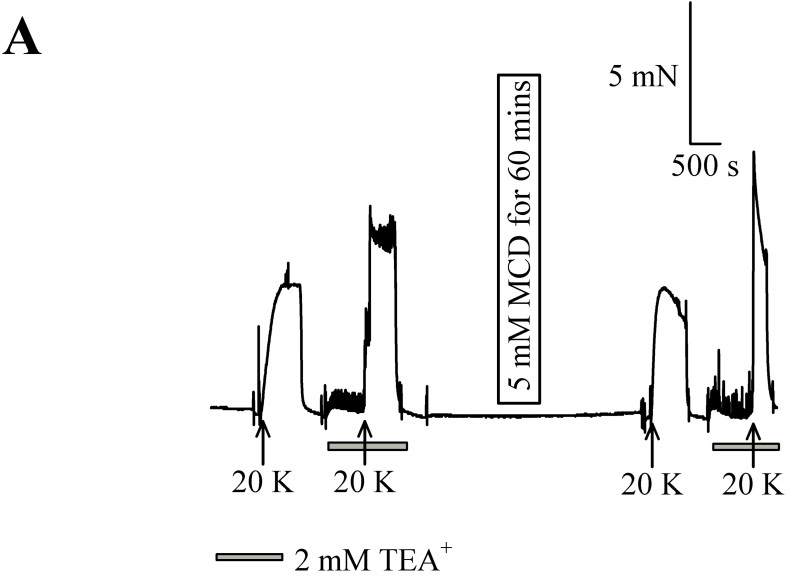

**B**

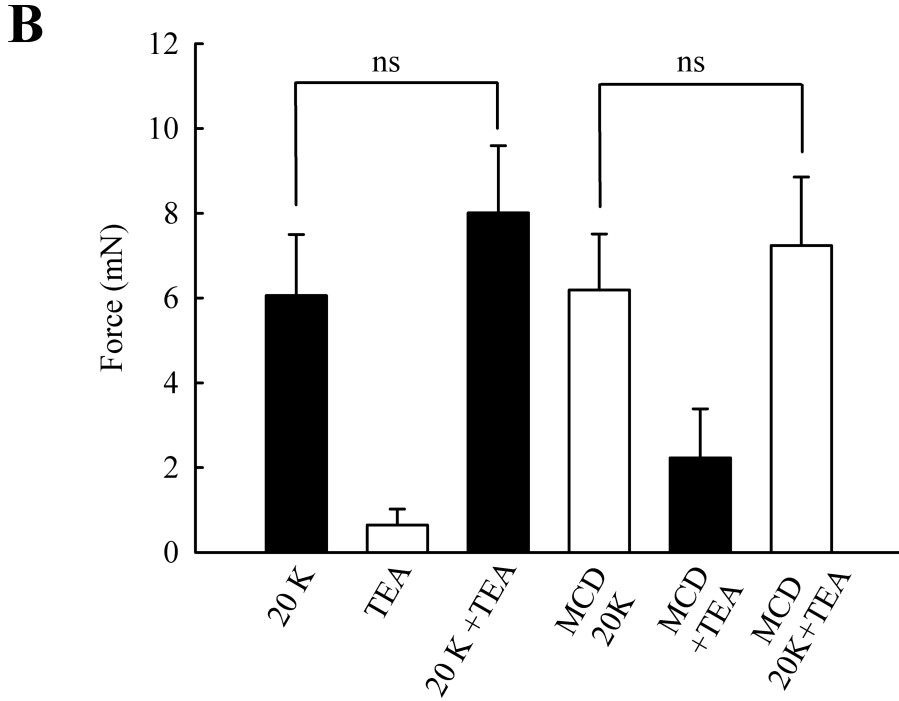

**Figure 8 Effect of TEA$^+$ on 20 K/Bay K contractions in an endothelium-denuded artery.** (A) 2 mM TEA$^+$ had little effect on 20 K/Bay K contractions in endothelium-denuded artery. (B) Mean data showing a non-significant increase in the contraction response to 20 K/Bay K by TEA$^+$. Statistical significance was examined using ANOVA. $n = 8$.

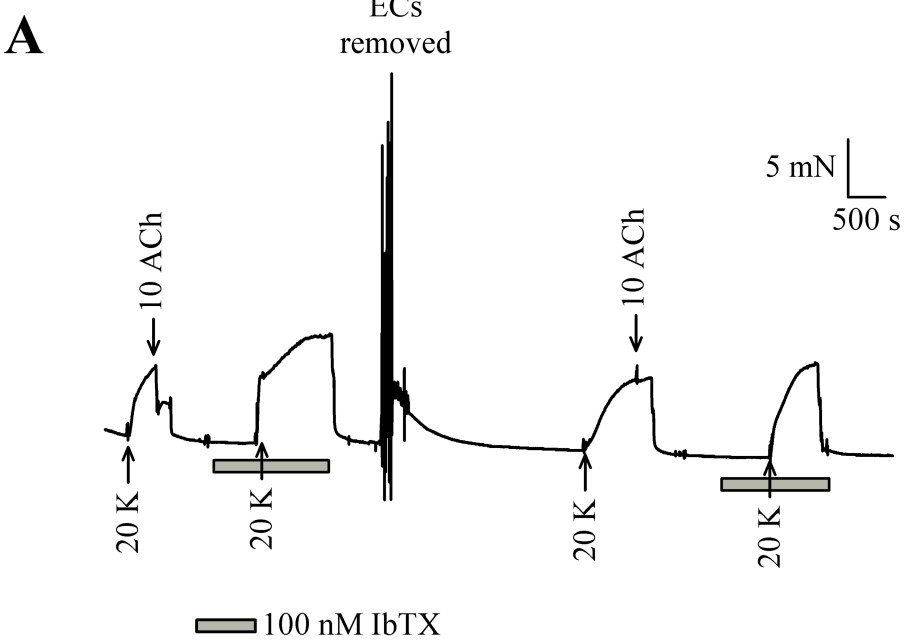

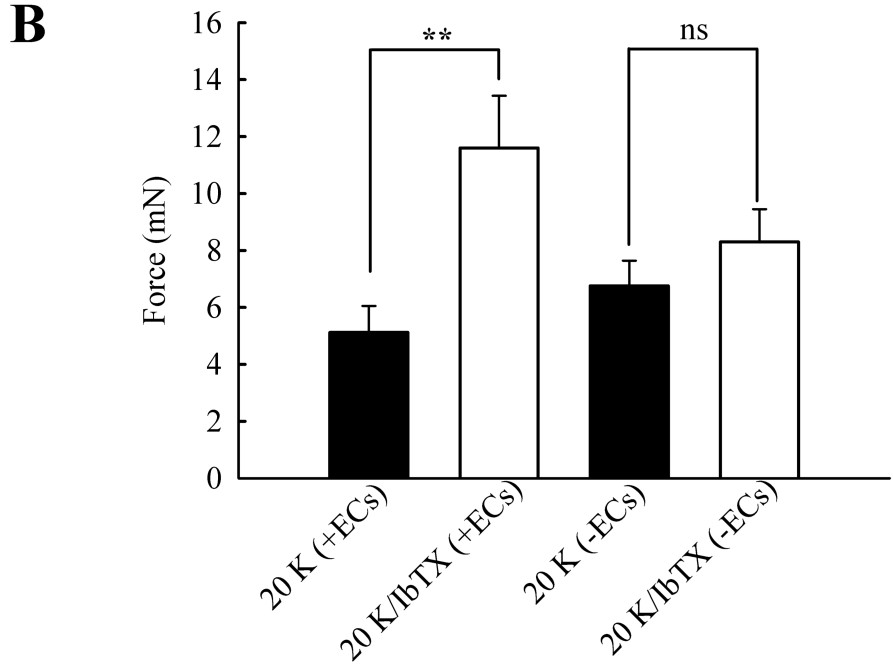

**Figure 9** **Effect of IbTX on 20 K/Bay K contractions.** (A) Original trace shows incubation of artery with 100 nM IbTX augments contraction in response to 20 K/ Bay K. After endothelial removal, IbTX was no longer effective at increasing force. (B) Mean data showing an increase in the contraction response to 20 K/Bay K after incubation of the artery with IbTX. Statistical significance was examined using Student's $t-test$. $n = 6$.

**A**

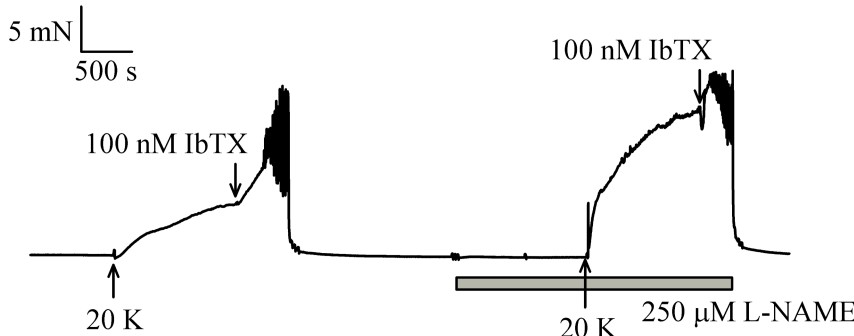

**B**

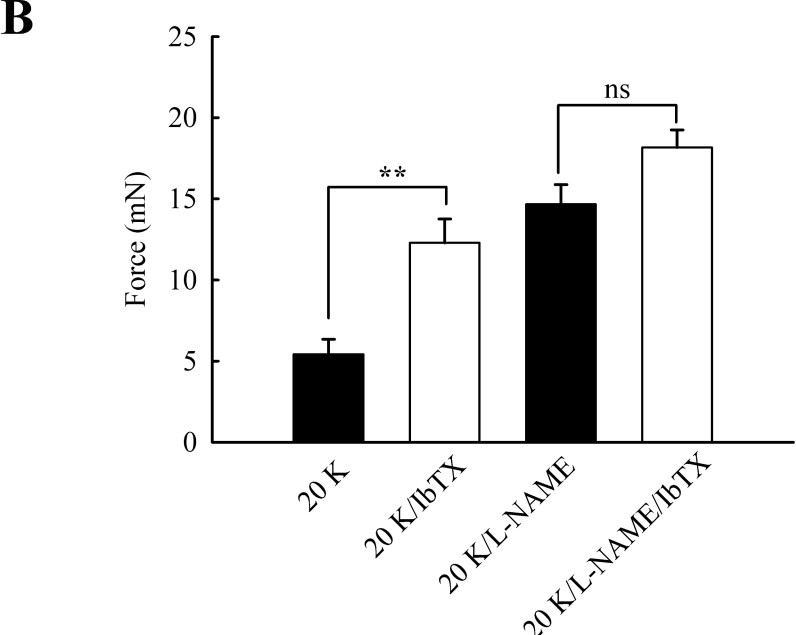

**Figure 10 Combined effect of IbTX and L-NAME on 20 K/Bay K contractions.** (A) Addition of 100 nM IbTX induced further contraction to 20 K/Bay K. When arteries were pre-incubated with 250 μM L-NAME, IbTX was less effective in increasing force. (B). Mean data showing a significant increase in contraction response to 20 K/Bay K by IbTX before, but not after, incubation with L-NAME. Statistical significance was established using Student's $t$-test. $n = 8$.

the contractile effect of IbTX was now much smaller. IbTX (100 nM) caused additional contraction over and above that caused by 20 K/Bay K of $6.86 \pm 0.81$ mN and $3.50 \pm 0.55$ mN ($n = 8$) in the absence and presence of L-NAME, respectively. Once again, this result is consistent with NO release from the endothelium causing activation of $BK_{Ca}$ channels in vascular SMCs, and that inhibiting NO synthesis with L-NAME or $BK_{Ca}$ channels with IbTX therefore has a similar overall effect on force.

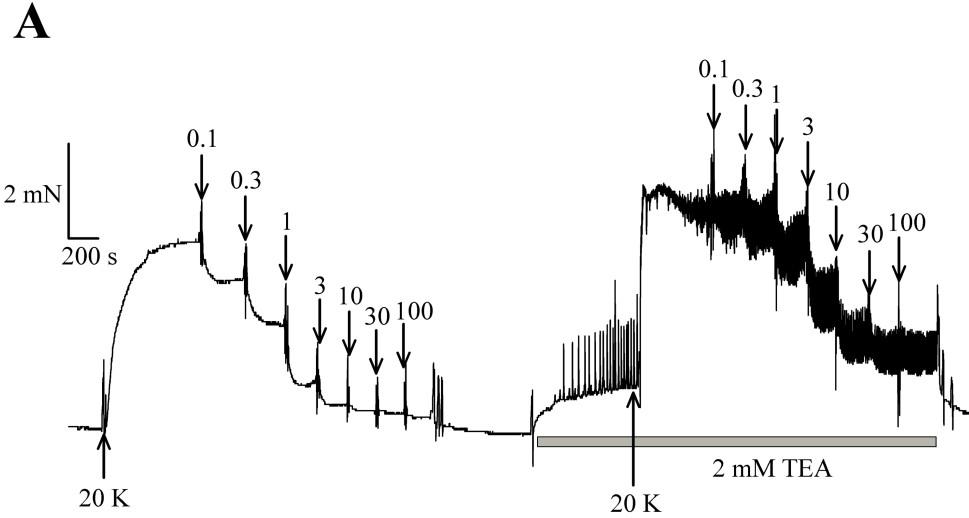

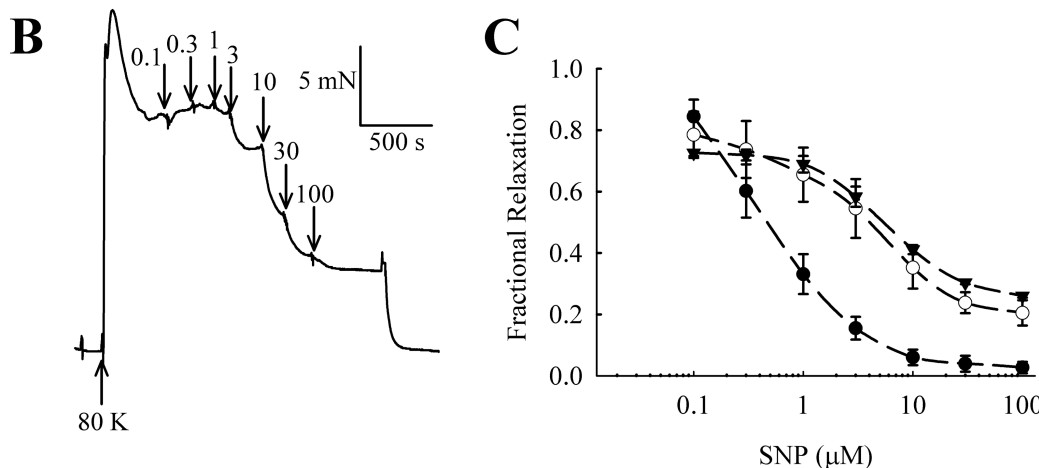

**Figure 11** **Effect of TEA$^+$ on SNP-induced relaxation in endothelium-denuded femoral artery.** (A) Traces showing vasorelaxation in an artery contracted by 20 K/Bay K. SNP (0.1–100 μM) produced relaxation before and after treatment with 2 mM TEA$^+$; (B) Trace showing vasorelaxation produced by SNP (0.1–100 μM) in an artery contracted with 80 K. (C) Concentration-response curves for relaxations induced by SNP in arteries contracted with 20 K/Bay K in the absence (●) and presence (○) of 2 mM TEA$^+$, and in arteries contracted with 80 K (▼).

## NO donors relax rat femoral arteries partly via activation of BK$_{Ca}$ channels

NO and other guanylate cyclase coupled vasodilators exert their effect in part through activation of BK$_{Ca}$ channels in some arteries (e.g., *Kahn, Higdon & Meisheri, 1998*; *Robertson et al., 1993*). In rat femoral arteries, the NO donor sodium nitroprusside (SNP) caused vasorelaxation, and this relaxation was attenuated in the presence of 2 mM TEA$^+$ (Fig. 11). The vasorelaxation response to SNP was significantly inhibited by TEA$^+$ at

concentrations of SNP of 1 µM and above (1 µM SNP, $P < 0.01$; 3–100 µM SNP, $P < 0.001$, $n = 8$). Furthermore, vasorelaxation in response to SNP was also attenuated in arteries contracted with 80 K solution (1 µM SNP, $P < 0.01$; 3–100 µM SNP, $P < 0.001$, $n = 8$). These results suggest that NO donors cause relaxation of rat femoral artery partly though activation of $BK_{Ca}$ channels.

## DISCUSSION

The caveolar disrupting agents M-$\beta$-CD and filipin both enhanced contraction of rat femoral arteries in response to 20 K/Bay K (Figs. 2 and 3). In the case of M-$\beta$-CD, the contraction was reversed by Ch-MCD (Fig. 4), and depended on an intact endothelium (Fig. 5). Arteries also showed enhanced contractions to 20 K/Bay K when NO synthesis was inhibited by L-NAME (Fig. 6). However, L-NAME did not augment 20 K/Bay K contractions following pre-treatment with M-$\beta$-CD. The $BK_{Ca}$ channel inhibitors $TEA^+$ and IbTX both enhanced contractions to 20 K/Bay K, and the effect of these agents was much reduced after removal of the endothelium or M-$\beta$-CD treatment (Figs. 7 and 8 and 9). Relaxations to the NO donor SNP were attenuated by inhibition of $BK_{Ca}$ channels with $TEA^+$ (Fig. 11). Taken together, these results suggest that caveolar disruption in rat femoral arteries results in a reduced NO release from ECs, and that this enhances arterial contraction due a reduced contribution of $BK_{Ca}$ channels to the smooth muscle cell membrane potential.

### Effect of caveolar disruption on endothelium-dependent vasodilation

Caveolae are specialised membrane domains characterised by a high cholesterol and sphingolipid content, and by the presence of the signature proteins the caveolins (*Rothberg et al., 1992*; *Cohen et al., 2004*). Caveolin-1 is necessary and sufficient to drive caveolar formation and knockout of caveolin-1 results in caveolar loss (*Drab et al., 2001*). Caveolin-2 is generally co-expressed with caveolin-1, with which it forms hetero-oligomeric complexes. Caveolin-2 appears to have a supporting role in caveolar formation (*Sowa et al., 2003*). Caveolin-3 is muscle specific and most muscle cells exclusively express caveolin-3, where it has roles analogous to caveolin-1 in non-muscle cells (*Cohen et al., 2004*). Vascular smooth muscle is unusual in expressing all three caveolin isoforms (*Cohen et al., 2004*; *Kamishima et al., 2007*). Caveolae have been shown to be important in physiological regulation of arterial tone, and disruption of caveolae may be a factor in vascular pathologies, including atherosclerosis (*Darblade et al., 2001*; *Xu et al., 2008*; *Prendergast et al., 2014*).

One particularly well characterised function of caveolae and the caveolins is in the regulation of the activity of endothelial nitric oxide synthase. Caveolin-1 binds to eNOS through a scaffolding domain, and this causes tonic inhibition of enzyme activity (*Michel et al., 1997a*). This inhibition is relieved in the presence of $Ca^{2+}$-calmodulin ($Ca^{2+}$-CaM), and endothelial $Ca^{2+}$ is therefore is a major regulator of NO synthesis (*Michel et al., 1997b*). Many factors stimulate NO production by increasing cytoplasmic $Ca^{2+}$, including endothelium-dependent vasodilators like acetylcholine (*Michel & Vanhoutte, 2010*). The functional effects of caveolin-1 loss have been addressed by gene knockout

(*Drab et al., 2001*). Arteries from caveolin-1 knockout (cav1-/-) mice have an enhanced response to the endothelium-dependent vasodilator acetylcholine, and have increased basal NO release (*Drab et al., 2001*; *Razani et al., 1990*). Both effects are consistent with the reported tonic inhibition of eNOS activity by caveolin-1.

The role of caveolae in vascular function has also been addressed by pharmacological disruption of caveolae by cholesterol depleting agents such as M-$\beta$-CD and filipin (*Zidovetzki & Levitan, 2007*; *Rothberg et al., 1992*; this study). In contrast to the studies on caveolin knockout animals discussed above, many studies have shown that disruption of caveolae by cholesterol depleting agents reduces the response to endothelium-dependent vasodilators such as acetylcholine (e.g., *Darblade et al., 2001*; *Graziani et al., 2004*; *Linder et al., 2005*; *Xu et al., 2007*). Caveolar disruption inhibits both NO-mediated and endothelial-derived hyperpolarizing factor (EDHF)-mediated vasodilation. In the case of NO-mediated vasodilation, disruption of caveoli by M-$\beta$-CD may result in (i) re-distribution of eNOS to non-caveolar membranes that contain caveolin-1, which then further inhibits eNOS activity, (ii) loss of local caveolar signalling pathways required to efficiently couple vasodilator-receptor binding to eNOS activation (*Xu et al., 2007*). It is possible similar mechanisms may be responsible for the effects we observed for M-$\beta$-CD in rat femoral artery. We have not specifically looked for EDHF-mediated responses in rat femoral artery. However, the NOS inhibitor L-NAME abolished the vasodilator response to ACh (Fig. 6), suggesting that NO is the main mediator of the response in these vessels. EDHF may become more important in resistance-sized arteries (see e.g., *Hill et al., 2010*; *Garland, Hiley & Dora, 2011*).

It should be noted that the agents we have used to deplete membrane cholesterol (M-$\beta$-CD, filipin) will also affect other (non-caveolar) lipid rafts (*Zidovetzki & Levitan, 2007*). However, given the well-documented localisation of eNOS within caveolae it seems more likely that our results can be understood in terms of the effects on this cellular structure.

Ch-MCD reversed the effects of M-$\beta$-CD, suggesting the effect was mediated by cholesterol depletion (Fig. 4). Interestingly, Ch-MCD significantly reduced contraction to 80 K solution. In this study we were focused on the effects of cholesterol depletion, and we have not investigated the basis for the effect of Ch-MCD further, but cholesterol has been shown to directly inhibit $K^+$-induced contractions of rat aorta by inhibiting voltage-dependent $Ca^{2+}$ currents (*Álvarez et al., 2010*). Cholesterol has also been reported to have a variety of direct, usually inhibitory, effects on ion channels (reviewed by *Levitan, Singh & Rosenhouse-Dantsker, 2014*).

## Direct effects of caveolar disruption on smooth muscle cell contraction

Disruption of caveolae in SMCs has previously been shown to modify the response to some receptor-coupled vasoconstrictors (e.g., *Dreja et al., 2002*; *Clarke, Ohanian & Ohanian, 2007*; *Potoknik et al., 2007*; *Prendergast et al., 2010*), and also reduces the myogenic response to increased intravascular pressure (e.g., *Adebiyi et al., 2007*; *Dubroca et al., 2007*; *Potoknik et al., 2007*). Generally, these results have been interpreted as providing evidence for the

importance of a caveolar location for receptors, signalling molecules or ion channels that are involved in the contractile response. In rat femoral artery caveolar depletion had relatively modest effects on contraction to 20 K/Bay K in the absence of the endothelium (Fig. 5), despite the presence of caveolae in SMCs (Fig. 1). Disruption of SMC caveolae may not have a major effect on contraction in these arteries.

## Basal NO release in rat femoral artery

Several studies have addressed the effect of caveolar disruption on relaxations to endothelium-dependent vasodilators such as acetylcholine (e.g., *Darblade et al., 2001*; *Xu et al., 2008*). Our study looked at the effect of caveolar disruption on NO mediated responses that occur in the absence of an endothelium-dependent receptor-coupled vasodilator. The eNOS inhibitor L-NAME increased contractions triggered by 20 K/Bay K (Fig. 6). PE concentration–response curves were also shifted to the right by pre-incubation of arteries in L-NAME (Fig. S1). Both observations are consistent with substantial basal NO release without an application of endothelium-dependent vasodilator in these arteries. Generally, NO synthesis by endothelial cells is low in the absence of a stimulating factor such as an endothelium-dependent vasodilator or flow. This is likely due to tonic and almost complete inhibition of eNOS activity by caveolin-1 in the absence of $Ca^{2+}$–CaM (*Michel & Vanhoutte, 2010*). However, basal NO release does occur and may be particularly important in large arteries (e.g., *Martin et al., 1986*; *Fleming et al., 1999*). The mechanisms of such a basal NO release have been debated and several hypotheses have been advanced. Lumenal flow can trigger NO release via Akt-induced phosphorylation and activation of eNOS, and this is likely to be important *in vivo* (reviewed in *Fleming, 2010*). However, our experiments were conducted in the absence of flow, so this seems unlikely to contribute to basal NO release in our study. Isometric contraction may by itself induce the activation of eNOS and so NO production (*Fleming et al., 1999*). Contraction is associated with a rise in vascular smooth muscle cell $Ca^{2+}$ concentration, and this can, in turn, bring about a rise in endothelial cell $Ca^{2+}$, so triggering $Ca^{2+}$-dependent activation of eNOS activity and subsequent NO release (*Dora, Doyle & Duling, 1997*; *Dora et al., 2000*; *Jackson et al., 2008*). $Ca^{2+}$ may move directly from smooth muscle to endothelial cells via myo-endothelial gap junctions (*Dora, Doyle & Duling, 1997*). Such a mechanism may contribute to the basal NO release observed in our study.

## Role of K$^+$ channels in the response to M-$\beta$-CD

Whilst contraction to 20 K/Bay K was enhanced by M-$\beta$-CD and filipin treatment, that to 80 K was unaltered (Figs. 2 and 3). When extracellular $K^+$ is raised to 80 mM, the membrane potential of the cell is depolarised and approaches the $K^+$ equilibrium potential, and so $K^+$ channel opening will not cause membrane potential hyperpolarisation or relaxation through closure of VDCCs. Thus if a reduced responsiveness to a vasodilator is seen as the extracellular $[K^+]$ is elevated this can be taken as evidence for the involvement of $K^+$ channels in smooth muscle cell in the response (e.g., *Meisheri, Cipkus Dubray & Oleynek, 1990*).

It should be noted that raising extracellular $K^+$ would also lower the driving force for $Ca^{2+}$ entry into the endothelial cell, which by itself may decrease basal NO release.

However, further experiments showed that 20 K/Bay K contractions were enhanced by TEA$^+$ and IbTX, both BK$_{Ca}$ channel inhibitors (Figs. 7, 9 and 10). This effect of TEA$^+$ and IbTX was lost if the endothelium was removed (Figs. 8 and 9). When arteries were pre-incubated in L-NAME the contractile effect of IbTX was considerably reduced (Fig. 10). The simplest explanation for our data is that endothelial-derived NO normally has a tonic inhibitory effect on femoral artery contraction via SMC BK$_{Ca}$ channel activation. Nitrovasodilators are known to cause vasorelaxation through K$^+$ channel opening. For instance, nitroglycerin becomes a less effective relaxant of dog coronary artery as extracellular K$^+$ is elevated, and the relaxations in physiological K$^+$ are reduced by pharmacological inhibition of BK$_{Ca}$ channels by charybdotoxin and iberiotoxin (*Kahn, Higdon & Meisheri, 1998*). The pathway for regulation of BK$_{Ca}$ channels by nitrovasodilators may involve activation of guanylate cyclase, generation of cGMP, activation of cGMP-dependent protein kinase and subsequent opeining of SMC BK$_{Ca}$ channels (*Robertson et al., 1993*). In rat femoral artery inhibition of BK$_{Ca}$ channels with TEA$^+$, and elevation [K$^+$]$_o$ to 80 mM, reduced the ability of the NO donor sodium nitroprusside to cause relaxation (Fig. 11).

### Endothelial BK$_{Ca}$ channels

An alternative explanation for the contractile effects of IbTx (Figs. 9 and 10) is that inhibition of BK$_{Ca}$ channels by this compound depolarises the endothelial cell membrane potential. In ECs the consensus view is that the presence of voltage-independent Ca$^{2+}$ 'leak' pathways means that membrane depolarisation will lead to a fall in Ca$^{2+}$ concentration due to an decrease in the driving force for Ca$^{2+}$ entry into the cell (reviewed by *Dora & Garland, 2013*, though see *Bossu et al., 1992*). This would reduce Ca$^{2+}$-dependent activation of eNOS, decrease NO release and so enhance arterial contraction. There are numerous reports of the presence of BK$_{Ca}$ channels in endothelial cells (reviewed in *Sandow & Grayson, 2009*). However, their presence is not universally accepted and it has been argued that they have mainly been reported in cultured cells and are not present in the endothelium of non-diseased arteries (*Sandow & Grayson, 2009*). In the context of the current study it is interesting to note that BK$_{Ca}$ channels in cultured bovine aortic endothelial cells (BAECs) cannot be recorded in resting conditions, but that they become active after cholesterol depletion (*Wang et al., 2005*). In BAECs, caveolin-1 appears to directly interact with and inhibit the activity of endothelial BK$_{Ca}$ channels. Interestingly, endothelial BK$_{Ca}$ channels activity can also be induced by activation of endothelial $\beta$-adrenoreceptors or by chronic hypoxia, and in both cases this occurs via release of caveolin-1 inhibition of channel activity (*Wang et al., 2005*; *Riddle, Hughes & Walker, 2011*). Further electrophysiological experiments will be required to determine whether BK$_{Ca}$ channels are present in rat femoral artery endothelial cells, or if their activity can be modulated by cholesterol depletion.

### Role of BK$_{Ca}$ channels in regulating femoral artery tone

In our study, inhibiting BK$_{Ca}$ channels with TEA$^+$ or IbTX enhanced 20 K/Bay K contractions in endothelium-intact arteries more than two-fold (Figs. 7, 9 and 10), but

only by about 25% in endothelium-denuded arteries (Figs. 8 and 9). Clearly $BK_{Ca}$ channels contribute to the membrane potential in both conditions, and the results are consistent with the well-documented role of $BK_{Ca}$ channels in providing negative feedback inhibition in conditions of tone generation (e.g., *Nelson et al., 1995*; *Hill et al., 2010*). However, the substantially larger contraction seen in the presence of an intact endothelium may mean endothelial NO is a major driver of $BK_{Ca}$ channel activity in rat femoral arteries. Interestingly $BK_{Ca}$ channels appear to make a relatively small contribution to membrane potential and resting tone in the arteries supplying the rat cremaster muscle, partly due to a low $Ca^{2+}$- sensitivity of the channel (*Jackson & Blair, 1998*). $BK_{Ca}$ channels form as a heterotetramer of pore forming $\alpha$ subunits, with the accessory $\beta$ subunits enhancing the $Ca^{2+}$-sensitivity of the channel. The low $Ca^{2+}$ sensitivity of the $BK_{Ca}$ channel in cremaster arterioles may derive from a relative lack of $\beta$-subunit expression in this artery (*Yang et al., 2009*). Our data suggests the contribution of the $BK_{Ca}$ channel to the membrane potential and contraction in femoral artery can be enhanced by endothelial derived NO, adding another factor that must be considered when assessing the role of this channel in the skeletal muscle vasculature. Regional variation in the regulation of $BK_{Ca}$ channels is likely to be important in fine control of vascular contractility in accordance with the physiological function of the tissue (*Hill et al., 2010*; *Yang et al., 2013*).

## CONCLUSIONS

Caveolar disruption in rat femoral artery results in contraction due to decreased release of endothelial-derived NO. The mechanism may involve a reduced contribution of $BK_{Ca}$ channels to the smooth muscle cell membrane potential. Endothelial-derived NO appears to be a major influence on the $BK_{Ca}$ activity in SMCs in these arteries, and this observation adds to our understanding of the complex regulation of $K_{Ca}$ channel activity in the skeletal vasculature.

## ACKNOWLEDGEMENTS

We would like to thank Simon Oliver and Alison Beckett from the Biomedical EM unit for their help and advice with electron microscopy.

### Funding

This work was funded by a PhD studentship awarded to Al-Brakati from the Saudi Arabia Ministry of Education. The funders had no role in study design, data collection and analysis, decision to publish, or preparation of the manuscript.

### Grant Disclosures

The following grant information was disclosed by the authors:
Saudi Arabia Ministry of Education.

### Competing Interests

The authors declare there are no competing interests.

## Author Contributions

- AY Al-Brakati conceived and designed the experiments, performed the experiments, analyzed the data, prepared figures and/or tables, reviewed drafts of the paper.
- T Kamishima conceived and designed the experiments, performed the experiments, analyzed the data, reviewed drafts of the paper.
- C Dart conceived and designed the experiments, reviewed drafts of the paper.
- JM Quayle conceived and designed the experiments, performed the experiments, analyzed the data, wrote the paper, prepared figures and/or tables, reviewed drafts of the paper.

## Animal Ethics

The following information was supplied relating to ethical approvals (i.e., approving body and any reference numbers):

Tissues were obtained from adult male Wistar rats (175–200 g; Charles River Laboratories) which were killed by a rising concentration of $CO_2$ followed by exsanguination. The care and euthanasia of animals conformed to the requirements of the UK Animals (Scientific Procedures) Act 1986.

## Supplemental Information

Supplemental information for this article can be found online at http://dx.doi.org/10.7717/peerj.966#supplemental-information.

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
