# Peer review of "Caveolar disruption causes contraction of rat femoral arteries via reduced basal NO release and subsequent closure of BKCa channels"

_PeerJ, doi:10.7717/peerj.966_

## Round 0.1 · original submission · Minor Revisions

The conclusions made in this study would be strengthened by adding further experimental data or expanding the discussion in regard to what mechanisms may be involved in basal NO release in rat femoral arteries, such as the contribution of passive tension or baseline blood flow.

·

Basic reporting

There are a few questions which can be raised because of slight differences between figures and related text:
Figure 2: Contraction levels after 20K/Bay are higher (almost x2) than in all the other experiments presented. This should be commented.
Figure 4: the effect of Ch-MCD on 80K showed in the figure is not commented.
Figure 7A: incubation with MCD is indicated but not discussed.

Experimental design

There are a few questions or comments which can be raised on experiments:

Figure 7: effect of LNAME on Phenylephrine is not really relevant if MCD action on Phe is not tested alone. This should be addressed.

Validity of the findings

No comments

·

Basic reporting

No comments

Experimental design

No comments

Validity of the findings

The Authors show that in rat femoral arteries there is a basal caveolae-dependent NO release, which opens BKCa channels and favours smooth muscle cells relaxation. Whereas this is a known mechanism, the Authors suggest that no other studies are available in the absence of an endothelium-dependent receptor-coupled vasodilator.
Indeed, in the work of Rees et al., 1990), mentioned by the authors, L-NAME alone induced a small but significant endothelium-dependent contraction of rings of rat aorta and, in the presence of phenylephrine, a greater endothelium-dependent contraction of the rings. These data suggest that a basal NO release do exist also in vitro.
Some experiments, or discussion, of the mechanisms operating basal NO release in rat femoral arteries should be performed. It should be excluded, or discussed, whether passive tension might trigger NO release. Furthermore, the dicothomy 80 K contraction versus 20 K/Bay K-induced contraction is not so clear. It may be supposed that NO release is fueled by a basal Ca2+ influx: if cells are depolarized by high K+, the driving force for Ca2+ is decreased and NO release reduced. Also the data with Iberio toxin should be discussed, since endothelial cells have BKCa channels and the toxin may depolarize these cells. Furthermore, the paper by Bossu et al (Pflugers Arch. 1992 Feb;420(2):200-7) should be mentioned, at least to evidence that no other reports of BK-activated channels in endothelium exist.

Additional comments

The Authors should focus on the problem of the mechanisms driving basal release, since a basal NO release has been evidenced and the pathway caveolae-NO- BKCa has been largely investigated.

---

## Round 0.2 · accepted · Accept

In the revised version, the authors have adress all of the reviewers comments and concerns, leading to a strengthened manuscript that is now acceptable for publication.